# Innovations in Sparkling Wine Production: A Review on the Sensory Aspects and the Consumer's Point of View

Maria Carla Cravero

CREA, Council for Agricultural Research and Economics, Research Centre for Viticulture and Enology, Via P. Micca 35, 14100 Asti, Italy; mariacarla.cravero@crea.gov.it

**Abstract:** Sparkling wines have a relevant economic value, and they are mostly produced world-wide with the Traditional method (in bottles) or with the Charmat method (in autoclaves). Many varieties are employed in different viticultural areas to obtain white or rosé wines and red (Italy and Australia), with different sugar content. This review illustrates the most recent studies (last 5 years) on sparkling wines concerning innovative yeasts, aromatic profile, aging on lees, sugar types, base wine, new varieties, and innovative oenological techniques, which consider the effects on the sensory characteristics and the consumer preferences.

**Keywords:** sensory analysis; sparkling wines; consumers; Traditional method; Charmat method

## 1. Sparkling Wines

Sparkling wines are a very important category of wines with a high economic value in different wine-growing areas worldwide. The global sparkling wine market reached a value of USD 42.12 billion in 2022 [1]. According to OIV data [2], in 2018, the world's sparkling wine production reached 20 mHL for the first time, with an overall increase of +57% since 2002, with an average increase of 3% per year. In the same year, the total value of sparkling wine exports reached a record of EUR 6.2 billion, which is 20% of the overall value of wine exported, and the global consumption reached 19 mHL [2]. The world production volume is concentrated (70–80%) in the European Union, especially in France, Italy, Germany, and Spain, followed by the USA. Other countries are less important. New producing countries have emerged in recent years, particularly the UK, Portugal, Brazil, and Australia. In the UK, sparkling wines represent more than 70% of the total domestic wine production [2].

In recent years, the trend observed for sparkling wines is a continuous increase and diversification; sparkling wine production has increased and shows no sign of reducing [1–3].

They are generally produced with a second fermentation in autoclaves (Charmat or Martinotti method) or bottles (Traditional or Champenoise method).

The excess pressure of carbon dioxide ($CO_2$) of endogenous origin in the bottle is at least 3.5 bars at 20 °C [2].

The "Champenoise method" should only be used officially for sparkling wines produced in the Champagne area in France (European Council regulation (EEC) N° 3309/85 of 18 November 1985 and EEC N° 2333/92 of 13 July 1992). The Traditional method has been widely studied for its aging potential and complexity. Even the glass is important for the best appreciation of these wines, and new shapes of glasses for tasting Champagne and sparkling wines were studied [4].

The so-called "Asti method" is a modified version of the Charmat process in which the grapes are harvested, crushed, and pressed. The must obtained is then filtered and fermented in pressurized stainless steel. There is not a second fermentation. In Brazil, sparkling wines produced by the Asti method are referred to as "Moscatel Espumante", and they should have an alcohol content between 7.0 and 10.0 mL/100 mL and a minimum residual sugar concentration of 20.0 g/L [5].

Another possibility to produce sparkling wines is Carbonation (i.e., a direct supplement of $CO_2$ into the base wine). In these wines, $CO_2$ is partially or totally of exogenous origin [2,6].

There is also the so-called Transfer method, a technique for making sparkling wine in which, after the second fermentation in the bottle and a short period of sur lie aging (but before riddling), the wine is transferred, with sediment, to a pressurized tank. The wine is then filtered under pressure and bottled [6].

White or red, aromatic, or non-aromatic varieties are employed to produce sparkling products with different sensory profiles. The products can be dry or with a sugar residue. Many factors influence the sensory profile of sparkling wines, such as the variety, the winemaking process, the aging on lees, and the yeasts [7].

In sparkling wines, the foam characteristics (foamability, persistence, in-mouth aggressiveness, and bubble size), the color, the aroma, and the acidity are relevant. This review considers the most recent studies (last 5 years) on sparkling wines concerning innovative yeasts, aromatic profile, aging on lees, sugar types, base wine, new varieties, and innovative oenological techniques, which consider the effects on the sensory characteristics and the consumer preferences (Table 1).

**Table 1.** The subjects and the references cited in the review.

| Subject | N° Citation |
|---|---|
| General information on sparkling wines | 1–7 |
| The effect of yeasts and inoculum | 8–19 |
| The volatile and sensory profile of sparkling wines | 20–29 |
| Aging on lees | 5, 7, 16, 30–35 |
| The effect of the sugar types | 36–40 |
| The effect of the base wine | 13, 42–45 |
| New varieties | 12, 46–53 |
| Innovative oenological techniques | 20, 48, 51, 55–59 |
| Consumers | 60–67 |

## 2. The Effect of Yeasts and Inoculum

The characteristics of sparkling wines, ethanol, carbon dioxide, mannoproteins, and precursors of aroma compound levels, and their foam and sensory profiles are influenced by yeasts [8]. *Saccharomyces cerevisiae* strains employed for the second fermentation have diverse flocculation degrees that can produce quantitative and qualitative different volatile profiles in sparkling wines [9].

Capozzi et al. [10] reported an overview of the production processes of sparkling wine, the main criteria for selecting *Saccharomyces* and non-*Saccharomyces* strains appropriate for the preparation of commercial starter cultures for the first and second fermentation, and the role of lactic acid bacteria. Moreover, the authors focused on the possible uses of selected indigenous strains.

The effect of non-*Saccharomyces* yeasts on sparkling wine sensory profiles is well illustrated in the 2018 review of Ivit and Kemp [11].

Eder and Rosa published another review in 2021 [12] that summarized relevant aspects of sparkling wine production using the Traditional method and non-conventional grape varieties and yeast starters for first and second fermentation. The authors argued that non-conventional grape varieties and novel or indigenous yeast starters are associated with the innovation and diversification of producing high-quality sparkling wines. These two elements can contribute to diversifying the sensory profiles of sparkling wines (Traditional method).

The non-*Saccharomyces* strains can exalt the sensory properties of the sparkling wines, even because there is a lack of diversity in the commercialized *Saccharomyces cerevisiae* yeasts employed in the second fermentation of sparkling wines [13]. These authors exploited the natural multiplicity of yeast populations to introduce variability in sparkling wines throughout the re-fermentation step. A collection of 133 *S. cerevisiae* strains was screened based on technological criteria (fermenting power and vigor, $SO_2$ tolerance, alcohol tolerance, and flocculence) and qualitative features (production of acetic acid, glycerol, and $H_2S$). These activities allowed the selection of some yeasts capable of dominating the in-bottle fermentation in actual cellar conditions compared to those of habitually used starter cultures. Results of chemical analyses and sensory evaluation of the samples after 18 months sur lies have shown that significant differences ($p < 0.05$) were present among the strains for alcoholic strength, carbon dioxide overpressure, and pleasantness.

In contrast, they were not observed for residual sugar content, titratable or volatile acidity. Indigenous *S. cerevisiae* exhibited values comparable to those of the commercial starter cultures. The authors concluded that the use of native yeast strains for the re-fermentation step can be considered a convenient way to introduce differentiation to the final product without modifying the traditional technology, and exploring yeast biodiversity can be a strategic activity to improve production.

A recent study by Tofalo et al. [14] on the impact of *Saccharomyces cerevisiae* and non-*Saccharomyces* yeasts showed that first sparkling wines—fermented with *S. cerevisiae*—were characterized by minerality, and they received intermediate scores for the other descriptors. Sparkling wines obtained with *Starmerella bacillaris* showed the lowest scores for all the descriptors considered. *Torulaspora delbrueckii* allowed obtaining sparkling wines with intermediate scores for some descriptors, e.g., "freshness", "bread crust", "fruity", and "minerality". Mixed fermentations showed the highest scores for the aromatic descriptors ("fruity", "floral", and "bread crust") in comparison with the sparkling wines inoculated with the pure cultures. The *S. cerevisiae* + *Starm. bacillaris* sparkling wine was significantly different from the others due to its high score for the descriptor "spicy", "bread crust", "freshness", and "floral". *S. cerevisiae* + *T. delbrueckii* sparkling wines showed the highest scores for fruity. The sensory analysis resulted in agreement with the differences observed in terms of aroma compounds since sparkling wines were obtained with *S. cerevisiae* + *Starm. bacillaris* were well differentiated from the others.

The same research group [15] investigated the influence of *S. cerevisiae* F6789A strain and its derivative mutants-harboring FLO1 gene deletion (F6789A-ΔFLO1) and FLO5 gene deletion (F6789A-ΔFLO5) on secondary fermentation, autolysis outcome, and aroma compounds production. The results revealed different metabolic behaviors leading to the production of sparkling wines with different characteristics. The floral and fruity attributes were more intense in sparkling wines fermented with the F6789A-ΔFLO5 strain, and they were less "herbaceous" than those obtained with the parental strain F6789A. These data agree with the analysis: the samples produced with the F6789A-ΔFLO5 strain had the highest content of isoamyl alcohol ("banana" odor), 1-butanol, 2-methyl ("truffle" odor), hexanoic acid, ethyl ester ("pineapple" odor), octanoic acid, and ethyl ester ("fruits" and "flowers" odors). The authors argue that these molecules are high-impact volatile compounds, in contrast with the opinion of Ubeda et al. [16], who wrote that the main molecules responsible for the "floral" attributes of sparkling wines were β-phenylethanol and diethyl succinate. The authors [15] concluded that considering the overall characteristics, all the strains produced balanced sparkling wines with negative and positive attributes arranged in good proportions, showing good aroma descriptors.

The wine aroma is the result of a high number of molecules, some volatile and some non-volatile, which can influence each other in a complex way, so it is not easy to attribute "flowers" and "fruits" odors to only a few compounds [17].

Employing a metabolomic approach, Tufariello et al. [18] very recently showed that autochthonous yeast strains can be an influential tool for innovation and market differentiation. The authors investigated the effects of four autochthonous yeast strains and

one commercial strain of *Saccharomyces cerevisiae* on the volatile and chemical profiles of rosé and sparkling wines (Bombino cultivar) produced in Southern Italy (Apulia region) with the Traditional method. The results suggest that the autochthonous yeast strains significantly influenced the composition of sparkling wines regarding volatile and non-volatile compounds. Moreover, a significant strain-specific effect of the autochthonous yeast strains on the aroma and metabolome of the sparkling wines was shown compared to the commercial strain.

The dry yeast preparation method for the second fermentation is also important. The study of Berbegal et al. [19] on the final "Cava" sparkling wines produced with the same base wine and inoculated with the same yeast strain showed that the aroma score was significantly higher in the samples inoculated with the pied de cuve-prepared yeasts than in the ones inoculated with the culture medium glucose, peptone, and yeast extract (GPY medium), even if the sensory analysis indicated no significant differences for the visual or flavor parameters. The trained panel (16 experts) evaluated the visual, aroma, and flavor characteristics using the score sheet for sparkling and pearl wines of the OIV. The intensity of each attribute was rated on a scale from 0 (low) to 10 (high). Higher significant values for total acidity, foamability, foam persistence, and content of total polysaccharides were observed in the wines inoculated with the GPY-grown yeast. These wines also had higher contents of esters, alcohols, and fatty acids with an Odor Activity Value OAV > 1.

### 3. The Volatile and Sensory Profile of Sparkling Wines

In a 2022 review, Romano et al. illustrated [20] the role of yeasts on the wine's volatile components. An interesting chapter of this review is dedicated to the effect of Non-Thermal Technologies on the production of wine aroma compounds. In particular, the authors illustrated some studies on the impact of high-pressure homogenization (HPH) on the volatile profiles of sparkling wines.

Numerous studies focused on the volatile fraction of sparkling wines produced with the Traditional method, even to test new products. De Souza Nascimento et al. [21] characterized volatile profiles of sparkling wines using Chenin Blanc and Syrah grapes, grown in the semi-arid region of São Francisco Valley (SFV) in Northeastern Brazil, where it is possible to have two harvests per year.

Muñoz-Redondo et al. [22] developed and validated a headspace solid-phase microextraction coupled with gas chromatography and mass spectrometry (HS-SPME-GC-MS) to determine 26 terpenes in 35 commercial sparkling wines from different grape varieties, geographical regions, and aging times.

The effect of pre-fermentative maceration and aging factors on the ester profile and marker determination of Pedro Ximenez sparkling wines was realized [23]. This is a Spanish sparkling wine produced with the low-aroma homonymous variety, generally used in the production of sweet wines and, along with the so-called "flor yeast" strains, responsible for the sensory characteristics of Sherry wine types. Martinez-Garcia et al. [24] analyzed the physicochemical and sensory differences of Pedro Ximenez sparkling wines produced with two yeasts aged on lees for 15 months. The results highlighted that aging affects the volatilome more than the yeast strain. PCA models obtained from odorant series and aroma descriptive tests showed good separation of the wine samples for both sensory and chemical data. In addition, the sensorial evaluation provided higher scores for the wines obtained with the native yeast at any aging time. Interesting associations between the content of volatile compounds and the odor attributes were observed, revealing a strong influence of thirty-eight compounds in the aroma.

In 2020, a Croatian survey [25] was carried out on the volatile aroma compounds profile and organic acid composition of commercial sparkling wine samples—produced with the Traditional method—from three vine-growing regions in Zagreb county.

The Romanian study of Cotea et al. [26] examined the effects of the type of inoculated yeast for the second fermentation on the composition of experimental sparkling wines produced with the Traditional method and obtained with the Muscat Ottonel aromatic

grape variety. The results showed a minor but important influence on the physical–chemical parameters. Regarding the sensory analysis and the volatile profiles, relevant differences were due to the type of the four inoculated commercial yeasts (V1—FIZZ™, V2—IOC DIVINE™, V3—LEVULIA CRISTAL™, and V4—IOC 18-2007™). Yeasts V1 and V3 showed the highest concentrations of aroma compounds, while V2 and V4 presented the lowest levels. Ethyl octanoate and ethyl decanoate were representatives for all yeasts, defining their "fruity" (especially "banana" and "apple") and "floral" notes ("elderflower").

Sparkling Prosecco is produced in Northern Italy (Veneto and Friuli Venezia Giulia regions) with the white variety Glera. Nowadays, it has vast commercial success all over the world, ranking first in the world among sparkling wines in terms of export volume (273 million liters), followed by Champagne (94 million liters) [27]. The aromatic characterization of 24 commercial Prosecco wines (price range EUR 7–13) was carried out in Italy [28]. The samples came from three different areas of origin: Valdobbiadene (15), Asolo (4), and Treviso (5). The wines were mainly characterized by yeast fermentation by-products (alcohols, acids, and esters) and C6-alcohols, followed by terpenes, low molecular weight volatile sulfur compounds (VSC), and benzenoids. The molecules with higher Odour Activity Values (OAV) were ethyl hexanoate ("fruit" odor), isoamyl acetate (banana odor), and β-damascenone (quince). Esters with fruity notes have an important role in the wine aroma. Some compounds discriminated the area of origin: the Asolo wines were richer in terpenes, norisoprenoids, and sulfur compounds; Valdobbiadene samples in benzenoids, while Treviso samples in hexanoic acid and phenylethyl alcohol. The sensory analysis realized with the sorting task method allowed to distinguish two main groups, not perfectly matching with the three areas considered.

Some Italian sparkling wines, 8 white, 3 rosé, and 4 red, were described in a study of characterization of 46 Italian wine samples applying the rate-all-that-apply method (RATA) with semi-trained judges by Rabitti et al. [29]. Recioto sparkling wine produced with the Traditional method was clearly distinguished by all wines being characterized mainly by the "caramel" odor and the macrocategories of "nuts" ("almond" and "hazelnut"), "dried fruits" ("figs", "prune", and "raisins"), "ethereal" ("flint stone" and "solvent"), and "vegetative" ("rosemary", "sage", and "marjoram"). The two Valdobbiadene DOCG and Moscato d'Asti wines were also clearly separated from all other wines. They were mainly characterized by the macrocategories of "floral" ("linden", "hawthorn", "acacia", "chamomile", "jasmine", and "orange blossom"), "caramelized" ("honey" and "vanilla"), and "fruit", including "tropical" ("pineapple", "litchi", and "melon"), "tree fruit" ("apple", "pear", "quince", "Moscato grape", "peach", and "apricot"), and "baked/ripe fruit". All the rosé wines and some white sparkling wines, including Alta Langa DOCG Extra Brut, Trento Metodo Classico DOCG, Franciacorta DOCG Brut, and Marche IGT, were similar in the mouth: "sour", "salty", and "astringent". Moreover, these three sparkling wines were characterized by similar odor attributes: "citrus" ("grapefruit" and "orange"), "earthy" ("mushroom" and "lees"), "vegetative" ("fresh-cut grass", "marjoram", and "sage"), and "yeast" ("yeast" and "bread crust").

## 4. Aging on Lees

The second fermentation of sparkling wines is followed by yeast autolysis and a period of aging on lees (composed by yeast, tartaric acid, and inorganic matter) [7]. Aging on lees can improve the quality of sparkling base wines [16,30]. The study of Ubeda et al. [16] carried out in Chile on sparkling wines produced with the Pais red grape variety with the Traditional method Blanc de noirs showed that the "fruity"/"floral" nuances could be due to a few high-impact aromatic compounds, (ethyl iso-butyrate, isoamyl acetate, ethyl hexanoate, β-phenylethanol, and diethyl succinate). Their content decreased during aging, but it was not perceived by the panel. The "bakery"/"toasty"/"yeast" nuances were perceived as more intense in the sparkling wines with higher aging time (9–12 months) compared to less-aged sparkling wines.

The autolysis process is very slow and expensive [31]. For this reason, many studies concern the use of yeast derivates in sparkling wine.

In the Spanish study of Ruiperéz et al. [31], the sensory characteristics and acceptability of enriched white Verdejo sparkling wines—added with β-glucanases or yeast derivatives (autolysated yeasts and yeast cell walls) in the tirage phase—were improved after a 22-month aging period. Total and neutral polysaccharide concentrations were higher in supplemented sparkling wines than in the control. Moreover, fruity and flowery characteristics were increased using yeast derivatives, while their smell of yeast character was higher with β-glucanases. Finally, the effect of these adjuvants was more intense in long-aged sparkling wines than in short-aged ones. Lambert-Royo et al. also studied the addition of yeast derivates [7]. In this case, the addition of 5 g/hL yeast protein extract and inactivated yeast from *Torulaspora delbrueckii* NSC19 helped to preserve esters responsible for "fruity" nuances in wines with 9 and 18 months of aging on lees. The addition of yeast autolysate achieved greater polysaccharide enrichment and a higher antioxidant activity. Sparkling wines treated with 10 g/hL of yeast autolysate and Optimum White™ generally exhibited the highest foamability and foam stability. Optimum White ™ is a commercial-specific inactivated dry yeast containing glutathione and polysaccharides. Further experiments with higher doses are needed to observe clear effects on sensory profiles.

The use of the ultrasound treatment of the lees, prior to the addition to the base wine, can improve the release of their components [30]. This recent study showed that sonicated lees reduced the astringency for increased neutral polysaccharides in the wine. Moreover, higher intensities of "floral "and "fruity" aromatic notes are due to the higher content of volatile compounds (acetates, esters, and terpenes). Also, some fused alcohols increased, contributing to the aromatic complexity of wines, especially 2-phenylethanol, alcohol with a rose-like aroma, and C6-alcohols with a green-herbaceous aroma.

The study of la Gatta et al. [32] on Bombino Traditional sparkling wine proposed an innovative technology to improve the wines' sensory profiles and reduce the aging period. Different volumes of lees recovered from the first fermentation were included in the "liqueur de tirage" for the second fermentation. The trained panel evaluated the following sensory characteristics: "effervescence", "olfactory intensity", "ripe fruitiness", "unripe fruitiness", notes of "yeast", "acidity", "structure", "body", "after taste", and "persistence". Adding lees up to 60 mL/L base wine positively influenced some sensory traits. In particular, the finesse and the complexity of wines increased, with higher intensities of "structure", "body", "aftertaste", and "persistence". A higher perception of delicate aromas like "unripe fruitiness" and notes of "yeast" and a consistent decrease for "ripe fruitiness", "overripe fruitiness", and "olfactory intensity" were observed.

The volatile composition of Moscato Giallo sparkling wines produced by Traditional, Charmat, and Asti methods was studied in Brazil [5]. All the methods significantly influenced the volatile composition of the final product. PCA analysis confirmed the effect of the production methods on the volatile composition of Moscato Giallo sparkling wines, which separated the samples into three distinct groups. The first principal component (58.85% explained variability) separated the samples produced with the Traditional and Charmat methods from those obtained with the Asti method. The second principal component (32.30% explained variability) separated the Traditional sparkling wines from the samples produced with the Charmat method. Traditional and Charmat method wines had the highest concentration of volatile compounds (ethyl octanoate, linalool, α-terpineol, 2-phenylethanol, hexanoic acid, and octanoic acid), and the sparkling wines produced by the Asti method showed the lowest concentration. The Traditional sparkling wines were correlated with citronellol, linalool, geraniol, hexanoic acid, octanoic acid, decanoic acid, isoamyl acetate, and ethyl octanoate, while the Charmat sparkling wines were correlated with hexan-1-ol and hotrienol. The Asti sparkling wines showed positive correlations with the oxide forms of linalool (oxide A (*trans*-furan), oxide B (*cis*-furan), and oxide D (*trans*-pyran)).

The higher alcohol 2-phenylethanol was the volatile compound found in the highest concentration in all sparkling wines (6118–8226 µg/L). The ethyl octanoate ester (pineapple flavor) had a high concentration in all sparkling wines analyzed, but mainly in the samples produced with the Traditional method (1229 µg/L), and concerning the terpenes, the highest concentrations were observed for linalool and α-terpineol and their content significantly contributed to the aromatic characteristics of the sparkling wines (Odor Activity Value OAV > 4). The authors did not carry out a sensory analysis of the wines. The Spanish study of Pons-Mercadé et al. [33] on Cava wines contradicted the expected release of polysaccharides and proteins due to lees autolysis. The authors wrote that these macromolecules are simultaneously released from lees and removed by precipitation, absorption, and/or enzymatic degradation. This complicated balance becomes even more complex given the inevitable variation in the original polysaccharide and protein fractions concentration between the different sparkling wine vintages. The authors adopted a new experimental approach that involved recovering lees from sparkling wines from nine consecutive vintages and reproducing the autolytic process in model wine. The panel of sparkling wine experts had to score the age of the sparkling wines from the nine consecutive vintages, from the youngest (1 point) to the oldest (9 points). The experts had to evaluate if the samples were "too old" by classifying them as "acceptable" or "not acceptable".

The results showed that both macromolecules are released from lees, but only sparkling wines aged for several years can benefit from increased polysaccharides and proteins deriving from yeast autolysis. The proportion of these compounds was low in the young sparkling wines: 2–3% in the first year of aging and ~7% after 3 years. The impact of these molecules in the sparkling wines disgorged before the end of the first year of bottle aging should be low.

The conclusions of this study are very interesting for wine producers since most Traditional sparkling wines are aged for less than 1 year, and those made by other methods are aged for an even shorter time. Moreover, the authors suggest that producers should consider that if the duration of aging is too long, sparkling wine quality can be negatively affected by excessive oxidation, as evidenced by the sensory results. The panel successfully differentiated the sparkling wines on the basis of their chronological production, except for some consecutive vintages. The sensory results indicated that after 6 years of bottle aging, the sparkling wines began to show excessive aging due to excessive oxidation.

Finally, contact with lees can increase the glutamate levels and total amount of free amino acids in aged Champagnes with long yeast contact and the sensory perception of the "umami" taste, as shown by Schmidt et al. [34].

Following the results of Franceschi et al. [35] on Italian still and sparkling white wines aged in contact with yeasts, the "umami" also plays the role of flavor enhancer and prolongs the aftertaste.

## 5. The Effect of the Sugar Types

The influence of different sugar types (glucose, fructose, and sucrose) in dosage at brut or demi sec residual sugar levels on consumer preference, aroma, and taste attributes was studied [36]. Although the wines were not aged following dosage addition, fructose and sucrose showed higher ratings ($p < 0.05$) for caramelized, vanilla, and honey aromas compared to glucose. Moreover, the results showed a consumer preference for wines sweetened with sucrose ($p < 0.05$) compared to glucose or fructose.

The increasing sucrose (cane sugar) levels in dosage (from 0 to 31 g/L) were associated with improved foam formation but reduced stability, possibly due to modifications of the wine's viscosity, as shown by Crumpton et al. [37] in English sparkling wines. The products were produced with a 50:50 Seyval–Chardonnay base wine blend. The different products were added to a solution of wine with the following granulated cane sugar contents: 0 g/L (Brut Nature), 5 g/L (Extra Brut), 10 g/L (Brut), 13 g/L (Extra Dry), and 31 g/L (Dry).

The Maillard reaction in sparkling wines produced with the Traditional method was recently studied by Charnock et al. [38]. The authors concluded that due to the conditions

during production and aging of these wines—low temperature (15 $\pm$ 3 °C) and low pH (pH 3–4)—the Maillard interactions may not proceed past intermediate stages, favoring the formation of furfural compounds (e.g., 5-HMF), which may be useful chemical markers during aging. The influence of specific Maillard-associated compounds on the aroma of sparkling wines is unclear and requires further sensory studies.

In a recent paper, the same authors [39] showed new insights into the chemical composition of sparkling wines during aging. Maillard reaction-associated products were quantified by headspace solid-phase microextraction coupled to gas chromatography–mass spectrometry (HS-SPME-GC/MS), and precursors were measured by enzymatic assay and proton (1H) nuclear magnetic resonance (NMR) spectroscopy.

The cane sugar or beet sugar addition to Auxerrois base wines produced in Canada was studied [40], confirming that the type of sugar used for the second fermentation of sparkling wines can determine an effect on the wine's volatile content, but a slight impact on the chemical composition. Beet sugar increased some volatile compounds in the sparkling wines after the second fermentation, likely due to its fatty acids deriving from the manufacturing process, which determines a high concentration of linear fatty acid-derived esters compared to cane sugar. In particular, beet sugar increased 1-hexanol and 2-phenylethyl alcohol in wines, although only 2-phenylethyl alcohol ("rose" and "floral" odors) was above its odor threshold (14,000 µg/L, determined by Ferreira et al. [41] in a 10% ethanol/water solution with 7 g/L glycerol at pH 3.2). The study evaluated only 14 volatile compounds with a "fruity" odor, and the authors conclude that further investigations are needed, including a higher number of sugar types and volatile molecules, a period of aging on lees, and the wine sensory analysis.

## 6. The Effect of the Base Wine

The wine type added in dosage had a higher impact on volatile aroma compounds and sensory properties of sparkling wines produced with the Traditional method than sugar addition during aging, as shown by Sawyer et al. [42].

The importance of the base wine composition in the production of Traditional sparkling wines is also evidenced by Vigentini et al. [13]. Their results showed that the base wine formulation is a key factor significantly affecting parameters like alcoholic strength, volatile acidity, carbon dioxide overpressure, titratable acidity, and dry extract. Other results show the importance of the base wine composition in influencing the aroma profiles of sparkling wines [43]. The importance of the base wine was also shown in a study carried out in Argentina with the aromatic variety Torrontés Riojano by Eder et al. [44] to compare wines produced using two yeasts (*Saccharomyces strains* EC1118, *bayanus* C12, and IFI473I) in the second fermentation. The sensory properties of these sparkling wines seemed to depend on the properties of the base wine more than on the yeast strain.

Very recently, Cisilotto et al. [45] compared sparkling wines produced by the Traditional and Charmat methods using the same base wine, yeast strain, inoculum, and aged on the lees during the same periods. The base wine was a blend of Chardonnay (36%), Riesling Italic (30%), and Pinot Noir (34%) vinified in white. The sensory analysis (triangle tests and Quantitative Descriptive Analysis (QDA)) confirmed the absence of evident differences, as shown via the physicochemical and volatile compounds analyses. More panelists could identify differences in the first stages. As the aging time on the lees increases, fewer panelists could differentiate between the sparkling wines, especially after 16 and 22 months. It was observed that more than half of the panelists could not differentiate the samples in all stages. The authors concluded that the quality of the base wine used plays a key role in both methods. Moreover, they wrote that the method used for the second fermentation is not the determinant of the eventual differences currently associated with sparkling wine produced with the two procedures.

## 7. New Varieties

New varieties have been experimented to produce sparkling wines in recent years. There is a rising interest in the production of red sparkling wines, an alternative for production and marketing in Spain [46–48] and Brazil.

In Brazil, Sartor et al. [49] elaborated sparkling wines with the non-traditional varieties Villenave, Niagara, Manzoni, and Goethe, studying phenolic composition, browning index, and glutathione content during 18 months of aging "sur lies". The same group studied the effect of mannoproteins on the evolution of rosé sparkling wines produced with Merlot grapes during over-lees aging with the chemical characterization of polyphenols, organic acids, macro- and microelements using a combined analytical approach [50]. The red cultivar Syrah is used to produce sparkling wines in Australia and Brazil [12,51].

Muscat sparkling wines from the traditional Asti region (Italy) and the new geographical indication "IG Farroupilha" (Brazil) were analyzed by Marcon et al. [52]. The multivariate analysis of the aromatic profile compounds differentiates the Muscat sparkling wines from Brazil or Italy. The quantitative differences affected residual sugar, sulfite content, isoamyl acetate, hexyl acetate, limonene, rose oxide, linalool, and citronellol. The sensory analysis confirmed that Brazilian wines were less sweet with a freshener aroma, whereas Italian wines were more intense in aroma, complexity, and sweetness. These different wine styles are associated with terroir and technological processes and are produced for different markets.

A recent study by Luzzini et al. [53] evidenced the industrially produced Durello sparkling white wines' volatile odor profile. Durello is an emerging PDO (protected designation of origin) sparkling white wine (Charmat or Traditional method) from the Veneto region (North-West Italy), and it must be produced with at least 85% cultivar Durella and the remaining 15% from other varieties (Garganega, Pinot Grigio, Pinot Noir, and Chardonnay). Some identified compounds were varietal markers (1,4-cineole and a group of non-megastigmane norisoprenoid), and some others derived from the principal and secondary fermentations (Classico or Charmat) and form aging on lees. These last compounds were different according to production technique and aging. For example, a statistically significant increase of 1,4-cineole was observed in older products. Moreover, the Traditional method reduced the content of cineole terpene precursors. Wines were subjected to sensory evaluation using the sorting task methodology, showing a combined influence of the production method, but also age/vintage and winery. Methanethiol, esters, and 1,4-cineole contributed primarily to the odor profiles. The 1,4-cineole and 1,8-cineole odors were described with sensory descriptive analyses as "hay", "dried herbs", and "blackcurrant", in Australian Cabernet Sauvignon wines and may be potential markers of regional typicality of these wines, with an odor threshold of 0.63 µg/L in red wines [54]. The content of 1,4-cineole in Australian wines was also determined: in Cabernet Sauvignon wines, it was $0.6 \pm 0.3$ µg/L, significantly higher than in Shiraz ($0.07 \pm 0.04$ µg/L) and Pinot Noir ($0.2 \pm 0.2$ µg/L).

## 8. Innovative Oenological Techniques

Different oenological techniques were studied on Tempranillo red base sparkling wines [48]. Pre-fermentative cold maceration with dry ice and delestage with early harvest grapes; sugar reduction in must and partial dealcoholization of wine with mature grapes. The pre-fermentative cold maceration was carried out via the addition of dry ice pellets (3 mm) to the destemmed and crushed grapes. The dry ice was added in a quantity necessary to decrease the temperature to $5 \pm 2$ °C and to maintain this temperature for three days before the beginning of the alcoholic fermentation. The oenological techniques used showed lower differences in the volatile composition of the wines than the grape maturity and the aging time. Considering the volatile content and the foam characteristics, the base wines obtained using mature red grapes showed more positive characteristics than those obtained from early harvest grapes, with the exception of the excessive alcohol degree obtained for this type of wine. The reduction of sugar content in musts and the

partial dealcoholization of wines produced base wines with more suitable alcohol content, but the high cost of these processes does not justify their use. The samples produced with early harvest grapes showed higher vegetal notes and a lower fruity aroma than those obtained from mature grapes. However, the pre-fermentative cold maceration allows for obtaining wines with a similar volatile composition to red sparkling wines produced from mature grapes and with the best value in the foam instrumental and sensory descriptors. The pre-fermentative cold maceration with dry ice resulted in the best solution to produce a red base sparkling wine.

The impact of cold pre-fermentative maceration using refrigeration on the nutraceutical quality and color of red sparkling wines elaborated with the cultivar Syrah was tested [51]. The antioxidant capacity and the phenolic content were higher in the sparkling wines elaborated with maceration, which also promoted a more intense red and saturated color.

A recent study by de Souza et al. [55] evaluated and compared the compositions of volatile compounds and sensory properties of sparkling and traditional wines (with and without $SO_2$) produced from Greek grapes "Grechetto", "Greco bianco", and "Greco di Tufo".

Potassium metabisulphite (10 g/HL) was added to the destemmed and crushed grapes after fermentation (15 g/HL). Before bottling, after the separation of the lees, the $SO_2$ content was adjusted to 40 mg/L.

The results showed differences between $SO_2$-containing, $SO_2$-free, and sparkling wines, with different content of alcohols, esters, fatty acids, phenols, and sensory characteristics. The "Grechetto" wine, without $SO_2$, showed volatile and sensory properties compared to the samples with $SO_2$.

A Brazilian study by Cisilotto et al. [56] showed that ethanol and sulfur dioxide have a synergistic effect on yeasts; this can be the main root cause of the problems sometimes encountered at the beginning of the second fermentation of sparkling wines (no start, a long lag period, or slow fermentation). This negative effect of ethanol, sulfur dioxide, and ethanol/sulfur dioxide on yeasts depended on the dose.

The $CO_2$ overpressure released during this second fermentation has an important effect on yeast metabolism and the wine aroma profile [57]. This study of Martínez-García et al. compared wines with or without $CO_2$ pressure, evidencing differences in the volatile profiles in open or sealed bottles. $CO_2$ overpressure affected ethyl esters of organic acids content. The PCA of 15 selected minor compounds (mainly ethyl dodecanoate, ethyl tetradecanoate, hexyl acetate, ethyl butanoate, and ethyl isobutanoate) evidenced the differences.

The aim to increase the South African sparkling wine quality led Jolly et al. [58] to verify the idea that a cork closure instead of a crown cap closure during the second fermentation and maturation on yeast lees can modify the final product characteristics. Six pairs of wines from five vintages, closed with cork or crown cap, were studied. Some differences were evidenced: cork-closed wines showed lower pressure compared to crown-capped wines. Albeit still well within legal requirements, the infrared spectral data, and the polyphenol profile (gallic, caftaric, caffeic, and p-coumaric acids concentrations). The infrared spectral data were different, but the nature of these differences could not be clarified. Smaller bubbles and a longer aftertaste were described in cork-closed wines, which tended to lose $CO_2$ from the glass slower after being poured than the crown-capped products. The authors suggested that the producers wanting to change their wine style can use cork instead of the crown cap during the second fermentation, even if more studies are necessary to clarify the differences.

Two alternative methods (Ancestral and Single Tank Fermentation) to produce sparkling wines were compared to Traditional and Charmat methods in a recent French study by Dachery et al. [59]. In the Ancestral method, the wine is bottled with 24 g/L of sugar, the second fermentation is performed in bottles, and there is contact with lees for 6 months. The final product is a Brut wine (sugar content < 12 g/L). The Single Tank Fermentation is a variation of the Asti method to produce sparkling wines with lower sugar levels (about 10.5%

alcohol). The volatile and sensory results confirm the possibility of employing these two techniques to obtain good-quality wines. Their sensory profiles were characterized by the odor attributes "floral", "tropical fruits", and "citrus fruits".

## 9. Consumers

Consumers have a high preference for sparkling wines; in some cases, women prefer this category of wines. Antoce and Logigan [60] showed the particularities of Romanian wine consumers in 2021 (166 men and 97 women). It was observed that men tend to prefer dry red wines, dry white wines, semi-dry white, and semi-dry red wines, while the women tend to favor sparkling wines, together with semi-dry rose wines, semi-dry aromatic wines, dry red wines, and sweet white wines.

It was recently shown [61] that sparkling wines with medium acidity, alcohol, and sweetness are versatile for pairing with a wide range of Thai street foods, from sweet to savory and spicy dishes.

An Italian study by Vecchio et al. [62] showed that the production process impacts both the sensory profile of sparkling wines and consumer expectations. In particular, the hedonic ratings revealed that when tasting the products, both with no information on the production process and with such information, the consumers preferred the Charmat wines. On the contrary, when detailed information on the production methods was communicated to consumers without tasting, the two Champenoise wines were more appreciated.

The purpose of Lerro et al. [63] was to show the changing consumer attitudes of different generations towards the sparkling wine consumption behavior and preferences of a large sample of US consumers (1096). Consumption frequency between genders is not statistically different, and women generally prefer sparkling wines priced below USD 15. Baby Boomers showed the lowest sparkling wine consumption frequency. Furthermore, Generation X and Baby Boomers have the highest consumption frequency in the USD 15–19.99 price range, while Millennials are in the USD 10–14.99 price range.

There was an influence of the COVID-19 pandemic on wine consumption, as shown by the online survey by Wolf et al. [64] carried out between 29 April 2020 and 7 May 2020, with 944 consumers from Western US States (California, Washington, Idaho, Oregon, and Nevada). The results showed the differences among four wine-consuming generations—Generation Z, Millennials, Generation X, and Baby Boomers—in wine-purchasing behavior, the desirability of wine attributes when making a purchase decision, and the information sources used. Products, prices, distribution channels, and messaging are important to target each generation. The authors suggest a national survey to compare their results obtained in California and neighboring states with the opinion of wine consumers in the USA.

The differences between the Traditional and Charmat methods in some conditions are not evident to consumers. In a recent paper, previously cited [45], some authors wrote that the method used for the second fermentation is not the determinant of the eventual differences currently associated with sparkling wine produced with the Charmat method and the Traditional method. They observed that more than half of the panelists could not differentiate the samples in all stages. In their experience, they compared sparkling wines produced using the same base wine, yeast strain, and inoculum and aged on the lees during the same periods. Moreover, they concluded that the quality of the base wine is very important for the wine quality.

Australian consumer preference for domestic sparkling white wines was studied in 2017 by Culbert et al. [65]. Evaluation of sparkling wines by an expert panel confirmed that complexity was closely associated with wine quality. The wines with the most intense toasty, yeasty, and aged/developed notes showed the highest quality score. Carbonated and Charmat wines tended to be fruit-forward sparkling wines, generally considered lower quality. Consumer acceptance was unrelated to wine quality or production method: an AUD 10 Charmat wine was liked more than Traditional method wines (on average). Four consumer clusters, each with distinct sparkling wine preferences, were identified. The perceptions and preferences of Australian wine consumers towards different styles of

sparkling wine, including French Champagne and Australian sparkling white, red, and rosé wines, Moscato and Prosecco, were studied in 2020 by the same group of researchers [66]. These results are interesting for sparkling wine producers to better target their products and marketing to the specific needs and expectations of Australian consumers of different segments. The authors conducted an online survey of more than one thousand (1027) regular sparkling wine consumers, and they collected demographic information, sparkling wine perceptions, and preferences. Moreover, the attitudes to purchasing and consuming this wine type were considered. Using a model, consumers were segmented into three categories, defined "No Frills", "Aspirants", and "Enthusiasts". The majority of No-Frills were females consuming sparkling wine only once per month. Almost 55% of Aspirants were male with a household income of more than AUD 75,000. Enthusiast consumers were also predominantly male and well educated, and 64% were under the age of 35 years. The preferred styles for each consumer group were sparkling white wine and Champagne, followed by Moscato and sparkling rosé wine. Moscato scored favorably with both the No Frills and Enthusiast segments. Almost 25% of participants declared to be not familiar with Prosecco, while sparkling red wine was perceived similarly by male and female consumers.

Three appellations—Champagne, Cava, and Prosecco—dominate the international market, while in the Italian market, there is a greater fragmentation due to the preponderance of many domestic products with a complex geographical classification system [67]. In this paper by Bassi et al., the Italian market of sparkling wines was investigated, particularly the repeated purchase behavior of sparkling wines in two years within the supermarket channel through scanner data collected from a consumer panel. The results showed a complex situation. Five segments were identified, and their evolution was followed over time: Ordinary Sweet, Ordinary Brut, Sophisticated, Prosecco, and Luxury. The loyalty to Prosecco changed strongly over time according to the region of residence, income, and family type.

## 10. Conclusions

This review[1] pointed out that the sparkling wine sector is continuously expanding and is the subject of study throughout all the wine-growing areas of the world. In the last 5 years, most of the authors consider important the sensory analysis of the products in their studies in addition to the definition of the volatile aromatic content.

Most of the papers concern Traditional sparkling wines and the second fermentation: yeasts, aging on lees, base wine, and sugar type. Innovative oenological techniques and the employment of new varieties—white, rosé, or red—are experimented with not only in emerging wine producer countries, like Brazil [2], but also in traditional wine countries (Spain and Italy).

In the papers examined, sensory analysis was generally employed to support the volatile aromatic composition data, describe the sensory quantitative wine profiles (QDA), or evaluate the quality of wines.

Innovative sensory methods have been pointed out in the last decades: flash profile [68], check-all-that-apply (CATA) [69], and rate-all-that-apply (RATA) [70,71]. They can be employed with consumers or with a trained panel, and they are more economical than the traditional QDA [72]. Only one paper of this review employed the RATA method [28], but in the future, it is hoped that these innovative sensory methods will be employed in studies on sparkling wines. The consumer studies literature mainly focuses on the preferences of sparkling wines independently from their sensory properties. Generally, other factors are taken into account, for example, age, gender, income, and wine information.

In the future, further studies should focus on sparkling wines with a low alcohol content or without alcohol, with a reduced addition of $SO_2$ and organic wines to approach the changing consumer attitudes. Advanced studies are also advisable on the diversification of sparkling wine production with new varieties, even rosé or red, or with autochthonous yeasts or non-*Saccharomyces* yeasts. The suggestion is to include the sensory analysis to evaluate the experimental wines.

Consumer preference linked to sensory properties should also be investigated to target better sparkling wines of different styles and economic value.

Lastly, a good subject to investigate would be the use of innovative and alternative containers and the evaluation of the wine shelf-life, including the sensory effects of some sparkling wines—produced with the Charmat method—for non-traditional consumers.

**Funding:** This research received no external funding.

**Conflicts of Interest:** The author declares no conflict of interest.

## Notes

[1] The literature references were searched in Vitis-vea.de data base https://www.vitis-vea.de (accessed on April–July 2023); science direct https://www.sciencedirect.com (accessed on April–July 2023); google scholar https://scholar.google.com (accessed on April–July 2023) using the terms: sparkling wine, sparkling wine and sensory analysis, sparkling wine and consumers.

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
