# Peer review of "Innovations in Sparkling Wine Production: A Review on the Sensory Aspects and the Consumer’s Point of View"

_beverages, doi:10.3390/beverages9030080_

Round 1
Reviewer 1 Report
The study focuses on an analysis of sparkling wines (mainly in a sensory analysis) in the last 5 years. The authors should mention at the outset that this sensory analysis is mainly focused on the aroma analysis of wines.
The article is well written but still lacks a little attention in its editing.
An explanation of how the literature reference search was done, what terms were searched, what databases were searched is also missing.
A list of points that should be improved are described below:
Page 2, Lines 56 and 62 and throughout the manuscript: instead of repeatedly mentioning "a described review", "a recent review", "another review", etc. The authors could for example write as “Capozzi et al. (2022) [6] suggested that…”
Page2, Line 76: Check the sentence: (Productions of acetic acid, glycerol and H2S)
Page 3, Line 102: S. cerevisiae in italics
Page 4, line 168: The following sentence is too long: "A Romanian study [21] on the volatile composition of experimental sparkling wines - produced by the traditional method - produced from the grape variety Muscat Ottonel, showed the effect of the type of pitched yeast for the second fermentation.” As well: Should "produce" mean "produced"?
Page 4, line 177: "Influence on physical-chemical parameters". What parameters? Or do the authors refer to physical-chemical conditions?
Page 4, line 197: Please add a space between "figs,prune".
Page 5, line 229: "Torulaspora delbrueckii" in italics
Page 7, 332: “Testing and Quantitative Descriptive Analysis (QDA))” maybe use square brackets but not double square brackets?
A critical point is why the study focuses mainly on samples/studies in Italy and Brazil. There is a reason these countries have been mentioned more frequently over the past five years (if so, it should be mentioned). Would other data from other regions like in France and Australia be used?
Page 7, 358: Reduce space: "Brazil or Italy"
Page 8, line 362: Check double spaces throughout the manuscript as: "Styles are linked"
Page 8, line 379: "Wines 41]": [41]
Page 9, line 427: "CO2" Please write the 2 as an underlined.
Page 9, line 441: "An Italian study [53]showed" add a space
Page 9, line 447: “The purpose of another work [54] was” Please change how works are cited throughout the manuscript. As already suggested: authors can refer directly to the name of their authors. For example: “Capozzi et al. (2022) [6] suggested that…”
Page 10, line 488: Review of the sentence: " and preferences, Moreover, the attitudes to purchase and consumption of this wine type were".
References: Check formatting: Lots of unformatted text like in Reference 35. Missing spaces and hyperlinks like in Reference 32. Authors should check the references carefully.
Reviewer 2 Report
This review is well organized and interesting for the production of sparkling wine. I suggest some minor points to improve the manuscript.
1. The content in some parts can be divided to several sections with subtitles according to the main idea, which might make it clearer.
2. Conclusions: some prospects or further research topics should be suggested.
3. Line 210: ‘ageing on less’ should be ‘ageing on lees’.
4. Change ‘aging’ to ‘ageing’ in the text.
5. Line 138: check the grammar.
1. Line 210: ‘ageing on less’ should be ‘ageing on lees’.
2. Change ‘aging’ to ‘ageing’ in the text.
3. Line 138: check the grammar.
Author Response
"Please see the attachment."

Reviewer 3 Report
Beverages-2542912 is a review of the scientific literature on sensory aspects of sparkling wine production innovations and consumer studies released in the last five years. Overall, the paper is interesting. However, I found the manuscript to be too much informative (i.e., a list of conclusions collected from other researchers), with little to no critical considerations typical of a review paper.
E.g., Authors A find AA, Authors B find BB, what’s the author explanation for these two different findings? This is missing throughout the manuscript.
I listed below other comments that should be addressed prior to publication.
L9: …they are mostly produced all around…
L10: (fermentation in bottles), (fermentation in autoclave)
L19-20: please, add an overall dollar value estimate (and specify if you are referring USD or AUD)
L26: …aging potential… on lees? Tank? Oak?
L30: …harvested, destemmed, crushed…
L35: …(i.e., direct supplement)…
L44: …aging on lees, the yeasts. “add a reference here.
L52: …yeasts [add reference]…
L97: when you mention sensory descriptors use “”. E.g., “spicy”, “bread crust”…here and elsewhere in the manuscript.
L115: So, what are the author's conclusions regarding these disparate findings?
L119: Instead of 4, use four. Numbers <10 are usually spelt out, while numbers >10 are written in digits.
L129: aroma score? Please, be more specific. What descriptor? How was it measured? Did they use trained panellists?
L159: aged for 15 months… where?? On lees?
L161: for both factors? Please, specify.
L184-185: please, use ‘yeast-fermentation by-products’ instead of ‘fermentation’
L196-198: please, use “ ” for every sensory descriptor mentioned here and elsewhere in the manuscript.
L210-283. This section needs some formatting work. It reads as one big paragraph.
L234: Optimum White> please, specify manufacturer and composition of this product.
L245: …sensory traits…? Which ones?
L252: volatile composition? Please, be more specific. What compounds caused the separation in three groups?
L264: …enzymatic degradation…reference is needed here.
L280-283: oxidation? What compounds and what sensory attributes were noted in this study?
L293-295: how was sucrose added? Grape concentrate rectified? Sugar cane?
L309: …an effect on the final product? What effect? Sensorial or chemical?
L315: specify threshold of 2-phenylethyl alcohol in wine (or ethanol/water solution).
L337: replace ‘evaluators’ with ‘panellists’.
L374: 1,4-cineole. Please, specify aroma descriptors of this compound and threshold in wine (or water/ethanol)
L379: when was dried ice used?
L384: please, replace ‘premature’ with ‘early harvest grapes’ here and elsewhere.
L389: cold maceration. Specify temp and length of the process.
L401: how much SO2 was used? And when was it added?
L415: just use PCA. It does not need to be spelt again.
L450: is $15 retail price? Specify if you are referring to USD.
L509-521: Conclusions. The authors should think about this section. What should the focus of future research be? What aspects have not been investigated?
Author Response
"Please see the attachment."

Round 2
Reviewer 3 Report
see attached

Author Response
Dear Reviewer,
thanks for your comments.
L301: trans-furan), oxide B (cis-furan) and oxide D (trans-pyran).
Please, change cis and trans to Italics
Ok I did this corrections.
L337: replace ‘evaluators’ with ‘panellists’. I used the same term of the authors of this paper.
I'm sure I'm misreading the author's point here, but nothing prevents the author of this review from
using a better term than the original if it makes more sense and is more grammatically accurate
(evaluators versus panellists). Evaluators are meaningless without context. What are the evaluators for? Panellists are better suited for sensory emphasis papers.
Ok, I changed the term evaluators with panellists in lines 397, 398, 400 and 556.
L30: …harvested, destemmed, crushed…Sorry, but the authors wrote harvested, crushed.
If the authors did not include 'destemmed' in their original article, they made a glaring error. It makes no sense to processing the grapes without destemming them
Ok, I did the correction.
I add Table 1 to summarize the subjects and the references cited in the review.
Other minor revisions:
Line 341 and line 343: I put 0.05 instead of .05
Line 370: odours instead of odour